# STNAGNN: Data-driven Spatio-temporal
# Brain Connectivity beyond FC

**Jiyao Wang**[1]                     JIYAO.WANG@YALE.EDU
**Nicha C. Dvornek** [1,2]
**Peiyu Duan** [1]
**Lawrence H. Staib** [1,2]
**Pamela Ventola** [3]
**James S. Duncan** [1,2,4]

[1] *Department of Biomedical Engineering, Yale University, USA*

[2] *Radiology & Biomedical Imaging, Yale School of Medicine, USA*

[3] *Child Study Center, Yale School of Medicine, USA*

[4] *Electrical Engineering, Yale University, USA*

**Editors:** Accepted for publication at MIDL 2025

## Abstract

In recent years, graph neural networks (GNNs) have been widely applied in the analysis of brain fMRI, yet defining the connectivity between ROIs remains a challenge in noisy fMRI data. Among all approaches, Functional Connectome (FC) is the most popular method. Computed by the correlation coefficients between ROI time series, FC is a powerful and computationally efficient way to estimate ROI connectivity. However, it is well known for neglecting structural connections and causality in ROI interactions. Also, FC becomes much more noisy in the short spatio-temporal sliding-window subsequences of fMRI. Effective Connectome (EC) is proposed as a directional alternative, but is difficult to accurately estimate. Furthermore, for optimal GNN performance, usually only a small percentage of the strongest connections are selected as sparse edges, resulting in oversimplification of complex brain connections. To tackle these challenges, we propose the Spatio-Temporal Node Attention Graph Neural Network (STNAGNN) as a data-driven alternative that combines sparse predefined FC with dense data-driven spatio-temporal connections, allowing for flexible and spatio-temporal learning of ROI interaction patterns. Our implementation is publicly available at https://github.com/Jiyao96/STNAGNN-fMRI/.

**Keywords:** Graph Neural Network, Functional MRI, Spatio-temporal learning

## 1. Introduction

Functional magnetic resonance imaging (fMRI) is a non-invasive imaging technique that measures brain activity by detecting changes in blood-oxygen-level-dependent (BOLD) signals. Through the use of fMRI, significant progress has been made in understanding the functional organization of the brain. Among the resting-state and task-based alternatives of fMRI, task-based fMRI presents more significant fluctuations in BOLD signal. It has been shown to be superior to resting-state data for applications such as predicting behavioral traits (Zhao et al., 2023) and detecting individual differences (Jiang et al., 2020). Although the diversity of task designs causes considerable difficulties in constructing large task-based

fMRI datasets, increasing evidence indicates promising potential for task-based fMRI. The task-induced fMRI signal may offer a strong inductive bias to learn an informative model, especially in studies where tasks are designed to enhance disease-specific brain activities.

In recent years, a wide range of machine learning methods including recurrent neural networks (RNNs) (Dvornek et al., 2017; Dakka et al., 2017), convolutional neural networks (CNNs) (Kawahara et al., 2017), and graph neural networks (GNNs) (Li et al., 2021a; Gadgil et al., 2020; Zhang et al., 2023) are applied to fMRI analysis. Among these approaches, GNN has its unique advantage in interpreting ROI-based brain interactions, an important field of research for understanding general brain functions and mechanisms of neurological disorders such as Autism Spectrum Disorder (ASD). However, efficient message passing and model interpretation in GNN rely on a high-quality definition of edges, posing considerable data processing challenges in the application of functional brain networks.

To utilize task-based fMRI data with both temporal task context and spatio-temporal ROI interactions, we formulate our goal as a spatio-temporal graph analysis problem. Specifically, we focus on discrete spatio-temporal graph formation where the spatio-temporal fMRI input is a temporal sequence of sliding window subsequences of the fMRI that we denote as graph snapshots. Although temporal GNNs have been a frequently studied subject in recent years (Rossi et al., 2020; Seo et al., 2018; Chen et al., 2018; Li et al., 2019; Pareja et al., 2019), we identify two key challenges unique to spatio-temporal brain graph applications:

- From the temporal dimension, the limited temporal resolution of fMRI acquisition and the sliding window truncation of the sequence data leads to a short sequence of graph snapshots, minimizing the advantages of the typical RNN to capture long-term dependencies in temporal information.

- For each graph snapshot, FC is more susceptible to noise when applied on short temporal sequences inside each sliding window. The noisy pre-defined edges are less likely to be accurate or sufficient in describing the brain's functional dynamics.

In the field of brain network analysis, methods incorporating multiple brain atlases (Wang et al., 2024) or multiple connectivity measurements (Chen and Zhang, 2023) are proposed to mitigate the challenges by introducing extra features in the input. In the field of graph theory, there are also attempts to detect and remove low-quality edges before training using geometric constraints (Park and Li, 2024). We propose to address these challenges from a data-driven perspective using STNAGNN, a spatio-temporal GNN model that incorporates a node-level attention algorithm for information aggregation on ROI-based brain graphs. To our knowledge, our approach is the first to implement direct spatio-temporal ROI connections at the node level, enabling more flexible information aggregation and model explainability. Meanwhile, it can also be applied as a complement to existing designs (Chen and Zhang, 2023; Wang et al., 2024; Park and Li, 2024).

## 2. Notation and Problem Definition

We truncate spatio-temporal fMRI data temporally into $T$ subsequences and construct each sliding window subsequence into a graph snapshot. For each instance, the input is a sequence of undirected weighted graph snapshots $\{G_1, G_2, \ldots, G_T\}$ where any $G_i = (V_i, E_i)$

is a graph in the vertex set $V_i$ and the edge set $E_i$. For any edge $(v_{i,j}, v_{i,k}) \in E_i$ connecting vertices $v_{i,j}$ and $v_{i,k}$, we define its edge weight $e_{i,j,k} \in R^+$. For a vertex set of $N$ vertices, $d$-dimensional input node features are denoted as $x_{i,j} \in R^d$ where $j \in \{0, 1, \ldots, N-1\}$.

Based on the above definitions, our goal of performing K class instance classification is equivalent to learning a mapping function $f$ that maps a sequence of graph snapshots to a class prediction label output $Z$:

$$f : \{G_i | i \in \{1, 2, \ldots, T\}\} \mapsto Z \in \{0, 1, \ldots, K-1\}$$

## 3. Data and Preprocessing

### 3.1. Biopoint Dataset

We include a 118-subject task-based fMRI dataset to experiment with an autism spectrum disorder (ASD) classification task. The dataset contains fMRI scans of 75 children with ASD and 43 healthy controls matched in age and IQ. The scans are acquired under the biopoint (Kaiser et al., 2010) task that contains 12 videos of biological or scrambled motions of point light displays. Videos of these two categories are given to subjects in an alternating sequence during the fMRI scan with the intention of highlighting deficits in the perception of biological motion in autistic children.

The scan for each subject has 146 frames with a frame rate of 2 seconds and an original resolution of 3.2mm. It is collected in the anonymous institution and approved by Yale Institutional Review Board. The acquired fMRI data are preprocessed using a pipeline described in (Yang et al., 2016), including the preprocessing steps of motion correction, interleaved slice timing correction, BET brain extraction, grand mean intensity normalization, spatial smoothing, and high-pass temporal filtering. The preprocessed data have a 2mm resolution in the MNI space.

### 3.2. Human Connectome Project (HCP) Dataset

We also evaluated our method in a 7-class brain state classification task using the HCP dataset (Van Essen et al., 2012). We take 1,025 subjects in the WU-Minn HCP 1200 subject data release who have RL task-based fMRI scans in all 7 fMRI task sessions: emotion, gambling, language, motor, relational, social, and working memory. We use preprocessed fMRI in MNI space with 2mm resolution. Models are trained to classify spatio-temporal graph inputs into their corresponding tasks during data acquisition.

### 3.3. Data Processing and Graph Construction

For our biopoint dataset, we first parcellate the brain fMRI data into 84 ROIs based on the Desikan-Killiany atlas (Desikan et al., 2006). For network training, we performed class-stratified sampling on subjects in five roughly equal length subsets for five-fold cross-validation. Then, the mean time series of each ROI is extracted using 1/3 of all voxels by bootstrap random sampling (Dvornek et al., 2018). We sample each ROI 30 times as data augmentation method, resulting in a total of $3540 = 118 \times 30$ instances.

For graph construction, we truncate the mean time series into 12 non-overlapping subsequences aligned with each video stimuli. Each subsequence has 12 or 13 frames. Using a

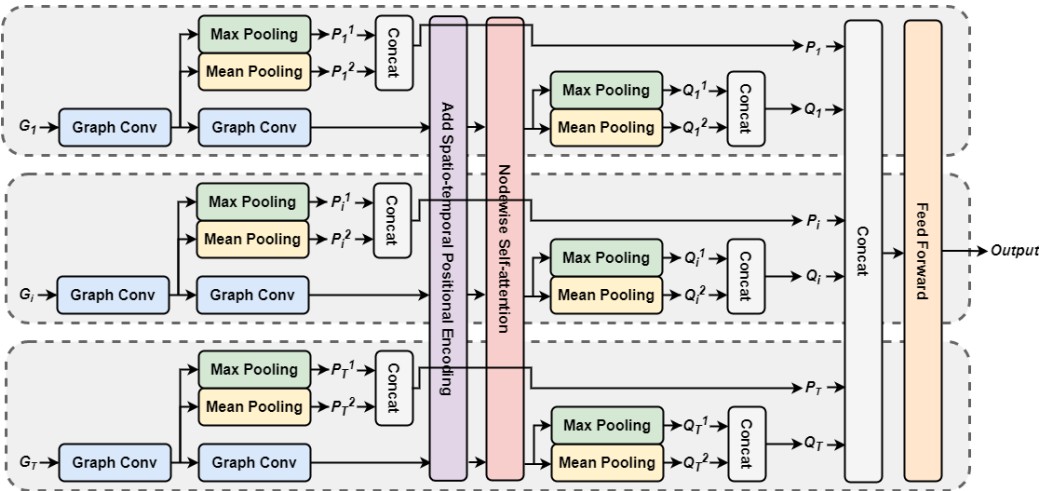

Figure 1: STNAGNN architecture

similar approach as described in (Li et al., 2021a), for each local subsequence time series, we calculate the Pearson correlation between ROIs and use it as node features. Meanwhile, we extract and concatenate all the time series acquired in biological motion videos. Using the concatenated sequence, we calculate a global biological partial correlation and use its top 5% values to define the edges and their weights for graph sparsity. Edges are shared across all 12 graph snapshots for each instance. As the biological motion viewing task is expected to elicit stronger correlated activity than the more random scrambled motion viewing task, we chose to use the global biological partial correlation for all edges. A graphical illustration of the preprocessing pipeline is shown in Figure 4.

For HCP data, we parcellate brain fMRI into 268 ROIs using Shen atlas (Shen et al., 2013). The 1025 subjects are also divided into five subsets of 205 subjects for cross-validation. Each subject has 7 scans for all tasks, which leads to a total of $7175 = 1025 \times 7$ instances. Data augmentation is not performed on the HCP dataset. For graph construction in HCP, we follow the same approach as biopoint data except truncating each HCP fMRI scan into 4 equal-length sliding window subsequences and calculating edges using each entire fMRI sequence.

## 4. Models

### 4.1. STNAGNN Model

Our proposed STNAGNN model utilizes GNN convolution operation and the attention algorithm for sparse-connection and dense-connection graph information aggregation, respectively. It maintains node identities in aggregating spatio-temporal information from different graph snapshots. As shown in Figure 1, after performing two layers of graph convolution to extract localized graph information, we add positional encoding to each node and compute nodewise self-attention as a global spatio-temporal information aggregation operation using the dot product attention algorithm (Vaswani et al., 2017). Essentially, in

this operation, we neglect the spatial edges defined by FC and impose a fully connected spatio-temporal graph containing nodes from all node sets $\{V_1, V_2, \ldots, V_T\}$.

Positional encoding is a crucial part for the attention algorithm to capture the order information of the data. In both Transformer (Vaswani et al., 2017) and Vision Transformer (Dosovitskiy et al., 2020), the additive positional encoding for the attention algorithm is an absolute 1D raster sequence sinusoidal function. Various other designs of positional encoding have also been proposed for transformer architectures, including relative (Dosovitskiy et al., 2020) and learnable (Li et al., 2021b) alternatives. There are also attempts to apply 2D positional encoding, but mainly in the application for x and y dimensions of 2D images (Raisi et al., 2021).

For graph-structured data, although adding positional encoding to nodes can potentially further empower GNNs with positional knowledge, additive absolute positional embedding is generally considered not applicable since it breaks the permutation invariance of graph message passing (Wang et al., 2022). However, for the ROI-based brain graph application, the brain graph nodes always follow the same node sequence. Permutation invariance is not a required attribute. In our proposed STNAGNN architecture, to encode both spatial and temporal information of a node and preserve computation simplicity, we propose an absolute multiplicative 2D positional encoding defined as follows:

$$PE(j, i, 2f) = sin(j/C_1^E)sin((C_2 + i)/C_1^E) \tag{1}$$

$$PE(j, i, 2f + 1) = cos(j/C_1^E)cos((C_2 + i)/C_1^E) \tag{2}$$

$$E = 2f/d \tag{3}$$

where $j$ denotes spatial position and $i$ denotes temporal position. $f$ represents the individual feature channel in the node features of dimension $d$. $C_1 = 10000$ is a constant to scale the frequency of encoding. $C_2$ is a constant offset to avoid duplicated embedding in different nodes. In our experiments, we set $C_2 = 10000$. When the spatial position $j$ is fixed, any positional encoding $PE_{j,i}$ can be represented as a linear function of $PE_{j,i+k}$ with $k$ being a constant temporal offset. The same applies to temporal position $i$ being fixed. As illustrated in Figure 1, the 2D positional encoding is added to the node features of each node before computing the self-attention operation. Meanwhile, since the proposed positional encoding can be pre-computed, the extra computation during training is negligible.

Our method is to our knowledge the first to use 2D positional encoding in a spatio-temporal context. There are several advantages in applying the spatio-temporal self-attention operation in the STNAGNN architecture.

- By imposing a fully connected self-attention operation, we mitigate the bias from inaccurate edge definition in the functional brain graph application. Data-driven information aggregation based on the similarity between node features and the spatio-temporal adjacency of nodes brings more flexibility to learning ROI interactions.

- A fully connected graph using attention allows for the direct participation of information from one node to any other nodes (Figure 2). It alleviates the problem of limited receptive fields in graph convolution operations (Kipf and Welling, 2016; Veličković et al., 2018). In our experiment, we argue that adding this operation allows multi-scale information aggregation from both local and global neighborhoods of nodes.

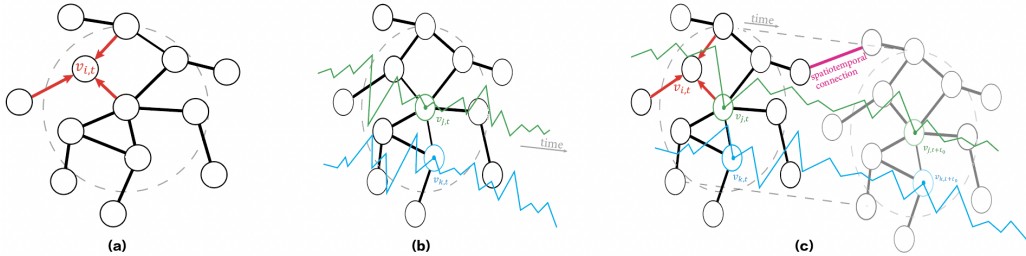

Figure 2: Illustration of connectivity types: a) spatial connectivity between neighboring nodes in one graph, usually computed by GNN convolution; b) temporal connectivity between different time points of one node; c) spatio-temporal connectivity (magenta), allowing information flow between nodes in different graphs. Existing architectures usually consider only spatial connectivity (Li et al., 2021a) or temporal connectivity (Dvornek et al., 2017). Some spatio-temporal designs consider both spatial and temporal perspectives (Seo et al., 2018; Chen et al., 2018; Li et al., 2019; Pareja et al., 2019) but use a two-step spatial-then-temporal approach. Our proposed STNAGNN jointly considers all spatio-temporal connectivity.

### 4.2. Baseline models

There are various existing spatio-temporal GNN models that use discrete graph snapshot structures to incorporate temporal information into GNN. For comparison with the proposed STNAGNN approach, we experiment with SVM and four different spatio-temporal GNN designs: GConvLSTM (Seo et al., 2018), GCLSTM (Chen et al., 2018), LRGCN (Li et al., 2019), and EvolveGCN (Pareja et al., 2019). These compared architectures include approaches of GNN-embedded RNN (Seo et al., 2018; Chen et al., 2018), stateful GNN (Li et al., 2019), and weight-evolving GNN (Pareja et al., 2019). In Section 5.2.3, we also show the performance of another implementation using LSTM instead of the proposed attention operation.

## 5. Evaluation and Interpretation

### 5.1. Classification Task Performance

For the STNAGNN model, we experiment with alternative two-layered graph convolution backbones including GCN (Kipf and Welling, 2016), GAT (Veličković et al., 2018), Graph-SAGE (Hamilton et al., 2017), and Graph Transformer (GT) (Shi et al., 2020). In the feed-forward modules following graph convolution and temporal aggregation methods, we apply SiLU (Elfwing et al., 2017) activation and a dropout rate of 0.2 in each layer. All models are trained with cross-entropy loss on a single RTX A5000 GPU. We perform a five-fold cross-validation experiment on both biopoint and HCP datasets. During training, we tune hyperparameters for each dataset respectively. For biopoint data, we use a learning rate of $2 \times 10^{-5}$ and a large weight decay factor of 0.015. For HCP, we use $4 \times 10^{-6}$ as learning rate and 0.0001 as weight decay. For both datasets, we use a batch size of 10.

|  | Biopoint | | HCP | |
| :---: | :---: | :---: | :---: | :---: |
|  | Acc(%) | AUC | Acc(%) | AUC |
| SVM | 68.7(4.97) | 0.608(0.046) | 91.2(0.991) | 0.993(0.001) |
| GConvLSTM | 63.8(3.65) | 0.642(0.031) | 96.7(0.768) | 0.998(0.001) |
| GCLSTM | 72.5(3.67) | 0.675(0.067) | 97.7(0.224) | 0.998(0.000) |
| LRGCN | 72.5(2.83) | 0.741(0.059) | 96.8(0.637) | 0.998(0.000) |
| EvolveGCN | 72.0(7.58) | 0.734(0.123) | 95.7(0.535) | 0.997(0.001) |
| STNAGNN-GCN | 75.2(4.40) | 0.670(0.130) | 97.3(0.424) | 0.998(0.000) |
| STNAGNN-GAT | **79.2(3.49)** | **0.755(0.099)** | 96.9(0.747) | 0.998(0.000) |
| STNAGNN-SAGE | 74.0(4.26) | 0.619(0.112) | **98.1(0.179)** | **0.999(0.000)** |
| STNAGNN-GT | 74.7(3.36) | 0.664(0.105) | **98.1(0.407)** | **0.999(0.000)** |

Table 1: Comparison of classification performance with SVM, temporal GNN baselines, and STNAGNN architectures using different graph convolution backbones. Results in biopoint and HCP dataset are reported in mean(standard deviation). Best mean performance in each column are bolded.

| Edge | ALL | | ALL | | ALL | | **BIOL** | | SCRAM | |
| :---: | :---: | :---: | :---: | :---: | :---: | :---: | :---: | :---: | :---: | :---: |
| # Windows | 10 | | 12 | | 14 | | **12** | | 12 | |
| GNN-backbone | GCN | GAT | GCN | GAT | GCN | GAT | **GCN** | **GAT** | GCN | GAT |
| Acc (%) | 73.4 | 73.5 | 73.4 | 73.5 | 69.7 | 71.6 | **75.2** | **79.2** | 70.0 | 73.0 |
| AUC | **0.680** | 0.666 | 0.621 | 0.665 | 0.658 | 0.710 | 0.670 | **0.755** | 0.676 | 0.666 |

Table 2: Ablation study on graph construction. BIOL, SCRAM, and ALL denote edge computed using fMRI under biological motion, scrambled motion, and all fMRI frames. The combination used above and the best performances are bolded.

The results for both datasets measured by classification accuracy and Area Under the ROC Curve (AUC) are summarized in Table 1.

## 5.2. Ablation Study

### 5.2.1. GRAPH CONSTRUCTION

We perform ablation studies of graph construction methods on biopoint ASD classification tasks for STNAGNN architecture using GCN and GAT. For data truncation, we validate the task-aligned choice of using 12 sliding windows by comparing it to using 10 and 14

|  | 1D Raster Sequence | | 2D Spatio-temporal | |
| :---: | :---: | :---: | :---: | :---: |
|  | Acc(%) | AUC | Acc(%) | AUC |
| GCN | 71.6 | 0.653 | 75.2 | 0.670 |
| GAT | 74.6 | 0.680 | 79.2 | 0.755 |

Table 3: Ablation study on positional encoding. Mean of cross-validation is reported.

| | Biopoint | | HCP | |
|---|---|---|---|---|
| | Acc (%) | AUC | Acc(%) | AUC |
| GCN-LSTM | 73.4(6.67) | 0.700(0.089) | 97.4(0.387) | 0.998(0.000) |
| GAT-LSTM | **76.0(5.31)** | 0.705(0.090) | 97.0(0.546) | 0.998(0.001) |
| SAGE-LSTM | 72.1(2.68) | **0.708(0.076)** | 97.8(0.471) | **0.999(0.000)** |
| GT-LSTM | 73.6(3.63) | 0.690(0.078) | **97.9(0.276)** | **0.999(0.000)** |

Table 4: Performance on GNN-LSTM ablation with various GNN backbones. Results in biopoint and HCP dataset are reported in mean(standard deviation). Best mean performance in each column are bolded.

windows using the whole sequence in edge construction for comparison. In addition, under the 12-sliding-window construction, we compare the performance of spatial graph edges constructed using fMRI data acquired under biological motion videos, scrambled motion videos, and the whole sequence. A visualization of the brain connectome using different graph construction methods is shown in Figure 5.The results are summarized in Table 2.

### 5.2.2. POSITIONAL ENCODING

We validate the proposed 2D spatio-temporal positional encoding by comparing it to the 1D raster sequence positional encoding. The results are summarized in Table 3. Using the proposed 2D spatio-temporal encoding outperforms using the 1D raster sequence option in all metrics. In Appendix B, we show a visualization of positional encoding methods to facilitate understanding of its interaction with the attention algorithm.

### 5.2.3. ATTENTION VERSUS LSTM

To compare the performance of the proposed attention-based spatio-temporal aggregation method with RNN structures such as LSTM (Hochreiter and Schmidhuber, 1997), we perform an ablation study on biopoint and HCP dataset using an architecture similar to STNAGNN except that the global attention operation is replaced with LSTM. Meanwhile, this modification also removes the global node-level connection and relies on LSTM to aggregate graph-level information in the temporal space. The architecture of this GNN-LSTM ablation is described in Figure 7. We experiment with the same set of GNN backbones as in STNAGNN architecture. The results in both classification tasks are shown in Table 4. The best performance in each column is not better than the best STNAGNN performance shown in Table 1.

### 5.3. ASD Biomarker Interpretation

An important advantage of graph-based methods in the analysis of brain fMRI is the capability to identify ROI biomarkers by interpreting the decision-making process of trained GNN models. To interpret our trained STNAGNN model, we apply the GNNExplainer (Ying et al., 2019), which is a module designed as a post-hoc interpretability method for GNN architectures. We consider all the $1008 = 84 \times 12$ spatio-temporal nodes and derive an

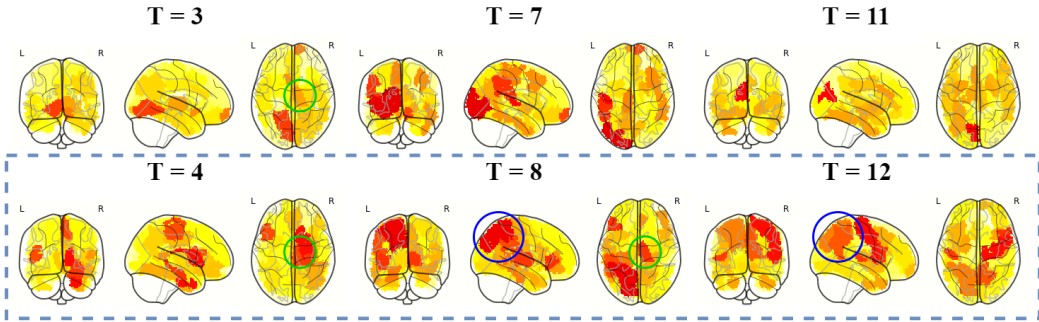

Figure 3: Interpreted ROI importance from $T = 3, 4, 7, 8, 11, 12$. Temporal indices of graph snapshots are marked on the top of each plot. Time under biological motion stimuli are marked by the blue dashed square. Darker regions indicate higher importance. Blue and green circles mark left parietal lobe and right thalamus. The complete plots of 12 sliding window snapshots are shown in Figure 8.

importance score for each node empirically by optimizing a mask function towards the highest mutual information between the outputs generated using masked and unmasked inputs. Using the trained best-performance STNAGNN with GAT backbone, we plot 12 ROI-importance heatmaps, each for a graph snapshot. 6 heat maps sampled among 12 snapshots of graphs are shown in Figure 3.

By comparing heat maps over different time frames, the brain ROIs that are important for making ASD classification appear to be dynamic across different graph snapshots. For example, from the heatmaps sampled, the signals from the left parietal lobe at T=8 contribute significantly more to the classification task than in the other frames. Meanwhile, we also see some recurring ROIs being prominent, such as the right thalamus in T=3, 4, 8 and the left parietal lobe in T=8, 12. While the thalamus is usually considered highly associated with ASD (Schuetze et al., 2016; Tomasi and Volkow, 2017), the left parietal lobe is also found to be indicative of language development in ASD (Zoccante et al., 2010).

Similar to other applications for interpreting ROI importance using GNN, analyzing recurring salient ROIs can help us identify potential ASD biomarkers useful for ASD diagnosis and subtype classification. Additionally, in a spatio-temporal GNN model such as STNAGNN, spatio-temporal importance aligned with task schemes can guide us in finding the appropriate stimuli to trigger the study-related functional response in the brain, which can potentially help design fMRI sessions.

## 6. Conclusion

In this paper, we propose the STNAGNN architecture as a spatio-temporal framework for analyzing task-based fMRI data. In addition to enabling spatio-temporal explainability, it also outperforms existing designs in both ASD classification and brain state classification tasks from two datasets. In future research, we intend to further develop this method, including experimenting with additional datasets and extending the tasks to predicting cognitive scores or performances in psychiatric assessments.

## Acknowledgments

This paper is supported by NIH grant R01NS035193.

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

# Appendix A. Related Work

## A.1. Graph Neural Network

Graph Neural Network (GNN) is a class of machine learning models applied to graph-structured data. It aims to learn an aggregation function for a neighborhood of nodes and propagate the function over the entire graph. In each layer, a GNN update of the node feature can usually be described as follows:

$$x_i^k = f(\{x_j^{k-1}, \forall j \in N(i)\}, x_i^{k-1}) \tag{4}$$

where $x$ denotes node feature, $k$ denotes layer number, and $N(i)$ is the neighborhood of node $i$. From a graph spectrum perspective, a GNN layer usually functions similarly to a low-pass filter on an input graph to suppress high-frequency noise and extract low-frequency information.

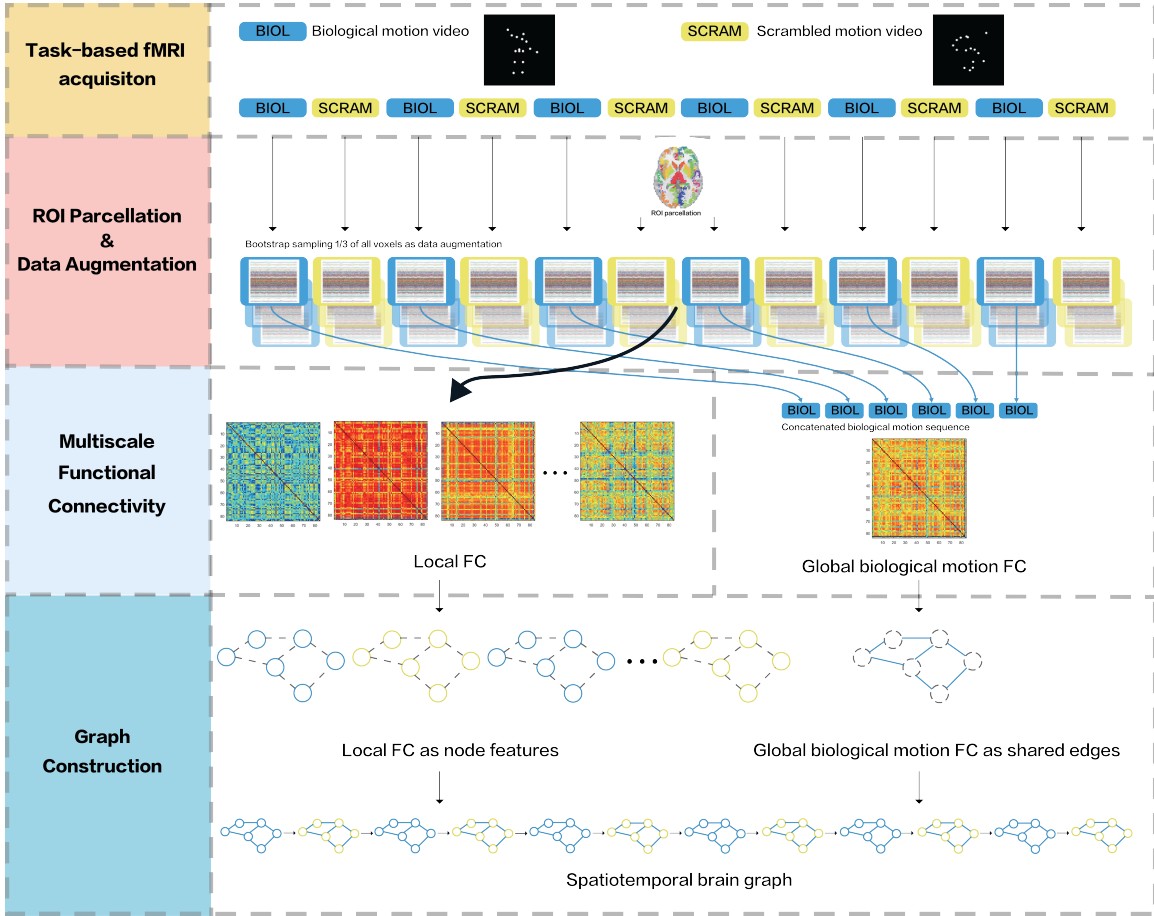

Figure 4: Preprocessing and graph construction pipeline on the biopoint data

## A.2. Scaled Dot-Product Attention Algorithm

The scaled dot-product attention algorithm is first introduced in ([Vaswani et al., 2017](#)) as a fundamental mechanism for the Transformer architecture. It quantifies the similarity between elements using regularized dot-product and updates each element as a weighted sum using the similarity-based attention score. A self-attention operation using this algorithm is theoretically similar to a fully connected graph, which allows for information to flow from one node to any other nodes.

$$Attention(Q, K, V) = softmax\left(\frac{QK^T}{\sqrt{d_k}}\right) V \tag{5}$$

## Appendix B. Visualization of positional encoding on simulated data

We used a simulated sample to visualize the interaction between different positional encoding methods and self-attention operation. We first generate a simulated setting with 84 nodes in each of the 12 temporal graphs so that it has the same shape as our Biopoint

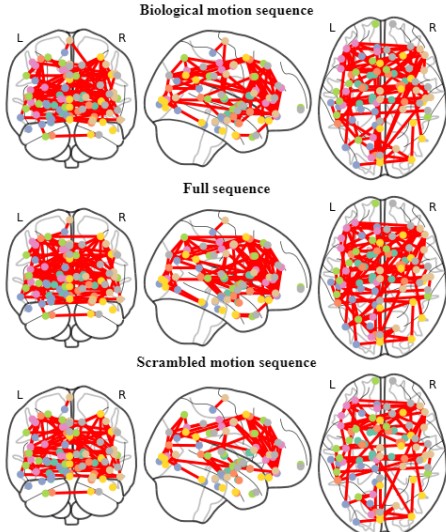

Figure 5: Visualization of calculated brain connectome using different approaches

data. For simplicity, each node feature is generated as a one-dimensional value following a normal distribution $\mathcal{N}(0, 0.1)$. We apply self-attention operation using three different approaches: 1) W/O positional encoding, 2) 1D raster sequence positional encoding, and 3) 2D spatio-temporal positional encoding. We extract the attention scores calculated during the self-attention operations, which directly explains how the node feature of each node participated in the attention operation output of each other node. We visualize the attention scores from one node as a heat map and compare its pattern under different positional encoding in Figure 6.

Without positional encoding, the attention score is entirely calculated from the randomly generated node features. Therefore, there is no noticeable connection between the attention weights and the underlying spatiotemporal structure. When 1D positional encoding is applied, it guides the attention weights towards an oscillation with pattern similar to sinusoidal function on the flattened raster sequence direction. Although it explains the affinity of nodes with similar node index in the spatial domain, it also shows obvious misalignment in the temporal space. The attention weights calculated from the proposed 2D spatiotemporal encoding shows a 2D pattern aligned with both spatial and temporal axis.

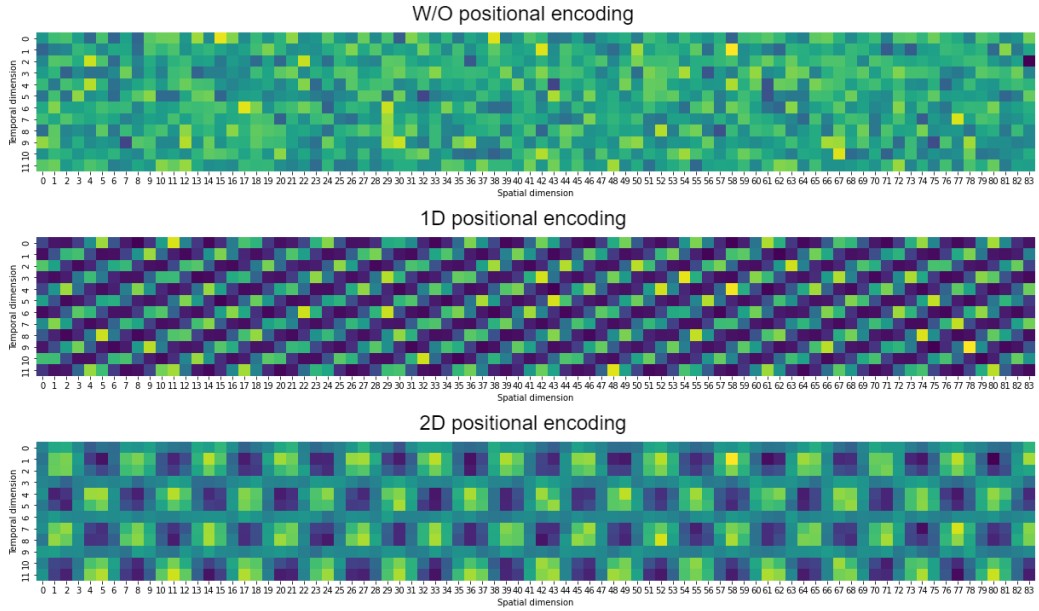

Figure 6: Visualization of the attention score calculated on simulated sample using different positional encoding method.

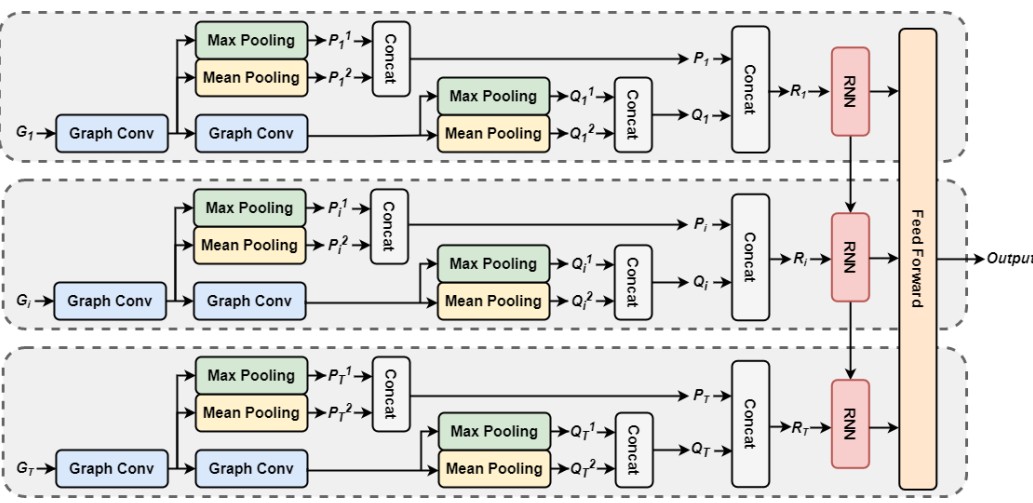

Figure 7: GNN-LSTM Architecture

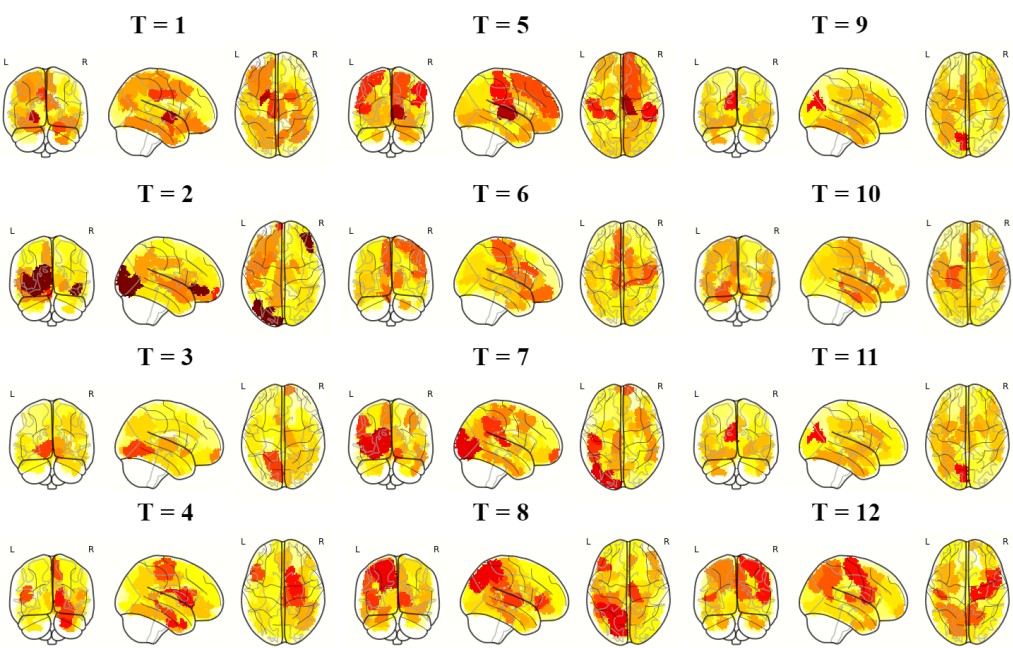

Figure 8: Interpreted ROI importance from $T = 1, 2, \ldots, 12$

