# OpenReview forum: "STNAGNN: Data-driven Spatio-temporal Brain Connectivity beyond FC"
_MIDL.io/2025/Conference — MIDL 2025 Oral_

### Official Review · Reviewer_FtVQ · 2025-02-13

**Confidence:** 4
**Preliminary Rating:** 4
**Recommendation:** Poster
**Final Rating:** 4

**Summary:**

The Spatio-Temporal Node Attention Graph Neural Network addresses challenges in analyzing task-based fMRI data by integrating sparse Functional Connectome (FC) with dense spatio-temporal connections. It employs 2D positional encoding and self-attention for flexible information aggregation and improved model explainability, tackling issues like noisy FC edges and limited receptive fields. STNAGNN shows superior performance in ASD classification and brain state tasks, demonstrated by higher accuracy and AUC values. The ablation studies validate the efficacy of the proposed methods and highlight potential ASD biomarkers, contributing to advancements in fMRI analysis and diagnosis.

**Strengths:**

1. The STNAGNN model integrates sparse predefined functional connectivity with dense spatio-temporal connections, overcoming the limitations of traditional methods. Its use of 2D positional encoding enhances the analysis of fMRI data, providing deeper insights into brain connectivity.
2. The paper employs two datasets—Biopoint for ASD classification and HCP for 7-class brain state classification—comprehensively evaluating STNAGNN against multiple baseline models. This thorough analysis validates the model's effectiveness and informs future improvements.
3. STNAGNN identifies dynamic ROI biomarkers for ASD classification through GNNExplainer, offering insights into neural mechanisms and practical implications for diagnosis and fMRI session design.
4. The paper addresses limitations in existing methods like functional connectivity and Effective Connectome, positioning STNAGNN as a solution. It provides a solid overview of related work, highlighting the model's significance.

**Weaknesses:**

1. The model relies on predefined-parcellation-based functional connectivity with known limitations.
2. Increased model complexity from 2D positional encoding and self-attention makes interpretation difficult.
3. The paper compares STNAGNN to only a few models, potentially missing relevant baselines. A broader literature review could provide a more comprehensive performance perspective.

**Detailed Comments:**

1.  It would be beneficial to include visualizations showing how different edge construction methods affect graph structure and the behavior of 2D positional encoding compared to 1D raster encoding. This can enhance the interpretability of the ablation results.

2.  An analysis of the STNAGNN model’s computational cost and memory requirements compared to baseline models would help clarify trade-offs between performance and resource consumption, including estimates of parameters, training/inference time, and memory usage.

3.  Even briefly exploring the model's application to other brain-related tasks, such as predicting cognitive scores or diagnosing neurological disorders, would broaden the paper’s scope and demonstrate its versatility.

4.  Details on the hyperparameter tuning process, including explored value ranges, tuning algorithms (e.g., grid or random search), and methods for determining optimal values, would aid reproducibility.

5.  A discussion on how components like 2D positional encoding influence self-attention and how graph convolution interacts with self-attention would provide deeper insights into the model's functioning.

**Justification Of The Final Rating:**

The authors have addressed all of my concerns and will potentially improve the manuscript significantly. In particular, the authors satisfactorily explain and supplement the necessary analyses. I have no more comments.

**Justification Of The Preliminary Rating:**

It offers a novel STNAGNN model that combines sparse FC with data-driven connections and uses 2D positional encoding. Experiments on two datasets with multiple baselines and ablation studies show its effectiveness. The model's ability to identify biomarkers is valuable. However, it has limitations like potential generalization issues and dependence on FC. Despite these, its innovative approach and strong experimental validation justify the high rating.

**Questions To Address In The Rebuttal:**

1. Authors should outline plans for testing STNAGNN's generalization beyond Biopoint and HCP datasets, including potential new datasets or reasons for broader applicability. Preliminary analyses showcasing adaptability to different data conditions would strengthen their claims.

2. Authors should suggest ways to reduce reliance on FC. They could explore alternative initialization methods and analyze the model’s sensitivity to FC by reducing its influence.

3. Authors should clarify how 2D positional encoding and self-attention interact within STNAGNN. Comparing a simplified model would help assess whether added complexity is necessary for performance.

4. Authors should justify their choice of baseline models and consider relevant alternatives. A literature review and sensitivity analysis of hyperparameters would validate their selections.

**Special Issue:**

No

---

> ### Author Response · Authors · 2025-03-07
> **Response to the comments from reviewer 3**
>
> We would like to thank reviewer 3 for a comprehensive review on our paper. We appreciate the opportunity to clarify and respond to each of the comments in the following section,
>
> 1. Visualizations showing different edge construction methods:
>
> We plot the connectome of a randomly selected subject in Figure 5 with edges constructed using the concatenated biological motion sequence, the full fMRI sequence, and the concatenated scrambled motion sequence. We threshold on edge weights by 0.15 so that only edges with stronger connection are shown. We observe that the edges calculated using scrambled motion sequence has a smaller amount of strong edges. We added this figure to the Appendix of manuscript.
>
> 2. Visualize 1D and 2D positional encoding and clarify how 2D positional encoding and self-attention interact within the STNAGNN:
>
> We used a simulated sample to visualize the interaction between different positional encoding methods and self-attention operation. We first generate a simulated setting with 84 nodes in each of the 12 temporal graphs using a node feature generated as a one-dimensional value following $\mathcal{N}(0,0.1)$. We apply self-attention operation using different approaches: 1) W/O positional encoding, 2) 1d raster encoding, and 3) 2d spatio-temporal encoding. We extract the attention scores of one node calculated during the self-attention and visualize as heatmaps in Figure 6.
>
> Without positional encoding, the attention scores show no connection to the underlying spatiotemporal structure. 1D positional encoding guides the attention weights towards an oscillation with pattern similar to sinusoidal function on 1D. Although it explains the affinity of nodes in the spatial domain, it shows misalignment in the temporal space. The attention weights calculated from 2D spatiotemporal encoding shows clearly aligned pattern in both spatial and temporal axis.
>
> In the STNAGNN architecture, as illustrated in Figure 1 of the manuscript, the 2D positional encoding is added to the node features before computing the self-attention operation. We added this clarification to Section 4.1.
>
> 3. Plans for testing STNAGNN's generalization beyond Biopoint and HCP datasets or classification tasks:
>
> In the table below, we show preliminary results on age prediction task in the Biopoint dataset using two best-performing baselines in classification tasks (GCLSTM, LRGCN) and STNAGNN using two GNN backbones (GAT, SAGE). The labels used for cross-validation training are standardized age values. We also added future plans to the conclusion section.
>
> |   | MSE | Correlation   |
> |:-|:-:|:-:|
> | GCLSTM | 0.018(0.003) | 0.226(0.126)|
> | LRGCN | 0.018(0.004) | 0.259(0.052)|
> | STNAGNN-GAT | **0.015(0.002)** |**0.346(0.119)**|
> | STNAGNN-SAGE | 0.016(0.003) | 0.300(0.081)|
>
> 4. Ways to reduce reliance on FC:
>
> We would like to clarify that although FC has known limitations, we still think it is an effective and efficient approach to model brain networks based on empirical evidence. We added literature review discussing works relevant to this issue in Section 1.
>
> 5. Comparing a simplified model:
>
> In Section 5.2.3 of the updated manuscript, we show an alternative implementation where the only difference from STNAGNN is using LSTM to aggregate temporal information instead of the proposed global attention operation.
>
> 6. Analysis of computational cost:
>
> We acknowledge that STNAGNN considers all ROIs as a global graph at the cost of memory. However, it does not affect the generalization capability of the model comparing to most of the existing methods. In the table below, we list  the computational resource consumption. STNAGNN allows us to implement spatial graph convolution in parallel, therefore showing improvement in running time.
>
> |   | GConvLSTM | GCLSTM | LRGCN | EvolveGCN | STNAGNN-GAT|
> |:-|:-:|:-:|:-:|:-:|:-:|
> |State dict | 9.7 MB | 8.5 MB | 7.9 MB | 7.0 MB | 12.3 MB |
> | GPU mem | 21.2 GB | 14.0 GB | 10.8 GB | 1.34 GB | 12.0 GB |
> | 1 epoch time | 208.2 s | 84.3 s | 73.7 s | 23.4 s | 12.2 s |
>
> 7. Justify the choice of baseline models and consider relevant alternatives:
>
> We select baselines using the same input as STNAGNN and thus function as alternatives. As mentioned in Section 1, designs may rely on additional inputs such as multiple atlases, joint FC and effective connectivity, or predefined temporal edges (Gadgil et al., 2020). It is difficult to compare with these models without introducing additional variables. Meanwhile, since STNAGNN can be combined with them. We do not consider it necessary to beat them in performance.
>
> 8. Hyperparameter search and reproducibility:
>
> The learning rate and the weight decay factor are determined by a grid search. The width and depth of layers are kept identical between different alternatives following an empirical optimum from previous experiments using GNN. To aid the reproducibility of our work, we have made our code publicly available as mentioned in the abstract of the updated manuscript.

---

> ### Comment · Area_Chair_2tQo · 2025-03-14
> **Please review responses and submit final rating**
>
> Dear Ye,
>
>     Could you please go through the rebuttal and indicate whether your concerns have been addressed adequately? Would you like to modify your rating in any way?
>
> The official discussion period ends today, so please acknowledge the response even if you intend to keep your original rating.
>
> -Your MIDL AC

---

### Official Review · Reviewer_xTjJ · 2025-02-19

**Confidence:** 3
**Preliminary Rating:** 5
**Recommendation:** Best Paper Award, Oral, Poster

**Summary:**

This paper proposes Spatio-Temporal Node Attention Graph Neural Network (STNAGNN), which combines sparse predefined FC with dense patio-temporal connections.  STNAGNN has node attention and a multiplicative 2D positional encoding, to model the interactions between different regions of interest in the brain. This model is trained and evaluated on two different datasets (tasks): Biopoint for ASD classification and HCP for brain state classification , and is shown to have better performance compared with previous GNN baselines. Results on different graph convolution backbones, graph constructions, positional encodings and model interpretation further showcase the effectiveness of this method.

**Strengths:**

1. The integration of spatio-temporal connections at the node level and the absolute multiplicative 2D positional encoding are the novelties of this paper, which are well-motivated and helpful to the model performance.
2. Experiments are comprehensive and showcase the robustness of the proposed method. Two datasets (Biopoint and HCP), one on ASD classification and the other on brain state classification are evaluated. Ablation studies on graph convolution, graph construction, and positional encoding provide thorough model analysis.
3. The overall presentation of the paper is clear and easy to follow. The brain plots highlight key brain regions (e.g., thalamus, parietal lobe) related to ASD, which enhances clinical relevance.

**Weaknesses:**

1. All evaluations are performed for task-based fMRI. If the paper claims a general modeling approach for brain connectivities, it would be better to show its applicability to resting-state fMRI.
2. Any ablations or results on using sparse predefined FC only? It can better showcase the effectiveness of integrating  dense data-driven spatio-temporal connections.
3. Any comparisons on non-GNN methods? e.g., CNN, RNN and Transformers?

**Detailed Comments:**

See weaknesses

**Justification Of The Preliminary Rating:**

This work with its novel idea on using node-level spatio-temporal connections with sparse predefined FC and 2D positional encoding, advances the application of GNN-based methods on brain connectivity modeling with clear presentation, well motivated and thorough experiments.

**Questions To Address In The Rebuttal:**

See weaknesses

---

> ### Author Response · Authors · 2025-03-06
> **Response to the comments from reviewer 2**
>
> We would like to thank reviewer 2 for pointing out additional baselines and ablations that can help strengthen our argument in this paper. We appreciate the opportunity to clarify and respond to each of the comments in the following section,
>
> 1. Application to resting-state data:
>
> From a methodological perspective, the proposed architecture can be applied to the resting-state data. However, theoretically, the proposed architecture is more appropriate for applications on fMRI signals containing diverse information in different graph snapshots, which is not guaranteed for resting-state acquisitions. Empirically, for ASD classification and biomarker identification tasks in the Biopoint data set, we consistently found that using task-based data over resting state data significantly improves performance in GNN-based structures using either BrainGNN (Li et al., 2021a) or the proposed STNAGNN model. We show an experiment in the table below that compares using resting state and task-based data with the STNAGNN-GAT model.
>
> | | Acc(%) | AUC   |
> |:-----------|:--------:|:--------:|
> | Resting-state| 71.6(4.64) | 0.641(0.133)|
> | Task-based   | 79.2(3.49) | 0.755(0.099)|
>
> 2. Ablation using sparse predefined FC only:
>
> In Section 5.2.3 of the updated manuscript, we show an ablation using LSTM to aggregate temporal information. In this ablation, the only connectivity at the node level is the sparse predefined FC. By comparing Table 1 and Table 4, the proposed global attention improves the performance of the model in both data sets.
>
> 3. Non-GNN baselines:
>
> We added SVM to the manuscript as a non-GNN baseline on both the Biopoint dataset and HCP dataset. The model in Section 5.2.3 can be considered as a baseline using both GNN and RNN. In the table below, we also experiment with a CNN model on the Biopoint dataset. Meanwhile, we would like to mention that analyzing fMRI images directly is memory-intensive. Using the image sequence as input consumes approximately 100 times more memory to load each instance compared to using the spatiotemporal graphs.
>
> | | Acc(%) | AUC   |
> |:-----------|:--------:|:--------:|
> | CNN | 67.8(5.47) | 0.714(0.043)|
> | STNAGNN-GAT | 79.2(3.49) | 0.755(0.099)|

---

> > ### Comment · Reviewer_xTjJ · 2025-03-11
> >
> > I thank the authors for their point to point response to my comments. I think with the additional experiments this paper is stronger. I will keep my original score.

---

### Official Review · Reviewer_NrFE · 2025-02-20

**Confidence:** 4
**Preliminary Rating:** 5
**Recommendation:** Oral
**Final Rating:** 5

**Summary:**

This paper proposes a novel method for fMRI data analysis based on graph neural networks. The proposed approach accounts for the spatio-temporal aspect of the data within a graph neural network through the use of positional encoding and nodewise self-attention. Results are presented on two different databases, compare the proposed method to SOTA temporal GNN methods, and evaluate the impact of the graph convolution backbone. The proposed method outperforms SOTA methods while allowing spatio-temporal explainability.

**Strengths:**

The paper is well written
Methods are detailed and the Appendix offers informative additional details
The method is novel, well designed, relies on strong prior studies and tackles a challenging task
The results are complete and several comparison methods were tested

**Weaknesses:**

The paper contains a lot of information some of which could benefit from a little more details, although I understand the space constraint. For instance in section 3.3 "Using a similar approach as described in (Li et al., 2021a)", it would be easier for the reader to have a sentence describing this method briefly.

**Detailed Comments:**

I believe Figure 2 could be improved to be easier to understand (what red arrows represent? Why are they directed? Dashed lines represent ROIs? on c) I am not sure to understand why these two nodes are connected with spatiotemporal connection)
3.3 - How many node features is there?
“we chose to use the global biological partial correlation for all edges”: How is that implemented at test time assuming labels are unknown?
Table 2 - it would be more informative to indicate the number of frames per window instead of windows.
Figure 6 is not referenced in the text

**Justification Of The Final Rating:**

Thank you to the authors for these clarifications and nice additional experiments answering all reviewers comments. I keep thinking that this revised paper will be of high interest for the MIDL community.

**Justification Of The Preliminary Rating:**

The methodology is novel and tackles an interesting problem, the methods are well validated on two databases and compared to SOTA. This work brings novelty to the field and is well written, justified and explained.

**Questions To Address In The Rebuttal:**

see detailed comments

**Special Issue:**

Yes

---

> ### Author Response · Authors · 2025-03-06
> **Response to the comments from reviewer 1**
>
> We would like to thank reviewer 1 for pointing out the confusions caused by our presentation of the paper. We appreciate the opportunity to clarify and respond to each of the comments in the following section,
>
> 1. Clarification of Figure 2:
>
> In Figure 2, each node represents one ROI. Each graph represents a temporal segment of the fMRI data. The dashed lines are drawn to show rough contours of the graphs and the direction of temporal space. The red arrows represent the spatial node feature update of one node computed by GNN. They are directed to show that the update of node features on each node is computed by the previous state of its own and all its neighbors. In part (c), the nodes are connected by spatiotemporal attention operation because the attention is computed on all nodes across all graphs (as shown in Figure 1 of the updated manuscript). We also added brief clarifications to the caption of Figure 2 in the updated manuscript.
>
> 2. Clarification on the number of node features:
>
> For each ROI, we compute its Pearson correlation with all the other nodes and use it as the node feature (Li et al., 2021a). Therefore, the node feature is a vector of dimension N, where N is the number of ROIs determined by the specific atlas. For the Biopoint dataset using Desikan-Killiany atlas, N=84. For the HCP dataset using Shen atlas, N=268. The number of ROIs and node construction methods are described in Section 3.3.
>
> 3. Clarification on using the global biological partial correlation as edges:
>
> In the Biopoint dataset, the fMRI acquisitions of all subjects are aligned with an alternating block task design of two categories of videos: biological point motion and scrambled point motion. The method of ``using global biological partial correlation for all edges" is to calculate the partial correlation between the ROI time sequences acquired during the biological point motion videos. In the ablation study shown in Table 2, we empirically found that this produces better performance than using the whole sequence or the sequence acquired under scrambled point motion. ASD classification labels are not required for this preprocessing. We include additional justifications for this approach in the responses to reviewer 3's comments.
>
> 4. Clarification of Figure 6:
>
> In Figure 3, we show 6 heatmaps of interpreted ROI importance from T=3,4,7,8,11,12. Figure 6 shows all the heatmaps from T=1 to 12. We added reference to Figure 6 in the caption of Figure 3 in the updated manuscript.
>
> 5. Details on number of frames in each window:
>
> We added details about the number of frames in Section 3.3.

---

### Author Rebuttal · Authors · 2025-03-06

**Rebuttal:**

We would like to thank all reviewers for their constructive comments on improving our paper! We attach the revised manuscript here with all modifications highlighted in red. For detailed explanations of all the modifications in bullet points, please refer to the official comments under each review.

**Supporting Material:**

/attachment/ddc786f9d5fd85892a2f4695d5ae271bdc66b814.pdf

---

### Meta-Review · Area_Chair_2tQo · 2025-03-20

**Recommendation:** Accept (Oral)
**Confidence:** 5

**Metareview:**

All reviewers are leaning towards accepting this work for publication at MIDL. It comes across as a nice solution and an interesting model design that addresses spatio-temporal modelling in fMRI data, which inherently challenging given the data-dimensionality and patient heterogeneity. Since the scores for the paper are quite favourable, this might be a good candidate for an oral presentation.